# Ecology and larval population dynamics of the primary malaria vector *Nyssorhynchus darlingi* in a high transmission setting dominated by fish farming in western Amazonian Brazil

**Paulo Rufalco-Moutinho**[1¤]*, **Samir Moura Kadri**[2], **Diego Peres Alonso**[2], **Marta Moreno**[3], **Gabriel Carrasco-Escobar**[4], **Catharine Prussing**[5,6], **Dionicia Gamboa**[7,8], **Joseph M. Vinetz**[4,8,9], **Maria Anice Mureb Sallum**[10], **Jan E. Conn**[5,6], **Paulo Eduardo Martins Ribolla**[1,2]

1 Departamento de Bioestatística, Biologia Vegetal, Parasitologia e Zoologia, Instituto de Biociências de Botucatu, Universidade Estadual Paulista, Botucatu, São Paulo, Brazil, 2 Instituto de Biotecnologia, Universidade Estadual Paulista, Botucatu, São Paulo, Brazil, 3 Department of Infection Biology, London School of Hygiene & Tropical Medicine, London, United Kingdom, 4 Laboratorio ICEMR-Amazonia, Laboratorios de Investigacion y Desarrollo, Facultad de Ciencias y Filosofia, Universidad Peruana Cayetano Heredia, Lima, Peru, 5 Department of Biomedical Sciences, School of Public Health, State University of New York-Albany, Albany, NY, United States of America, 6 New York State Department of Health, Wadsworth Center, Albany, NY, United States of America, 7 Facultad de Ciencias y Filosofía, Departamento de Ciencias Celulares y Moleculares, Universidad Peruana Cayetano Heredia, Lima, Peru, 8 Instituto de Medicinal Tropical "Alexander von Humboldt", Universidad Peruana Cayetano Heredia, Lima, Peru, 9 Department of Internal Medicine, Section of Infectious Diseases, Yale School of Medicine, New Haven, CT, United States of America, 10 Faculdade de Saúde Pública, Departamento de Epidemiologia, Universidade de São Paulo, São Paulo, Brazil

¤ Current address: Núcleo de Medicina Tropical, Universidade de Brasília, Distrito Federal, Brasília, Brazil
* paulorufalco@gmail.com

## Abstract

Vale do Rio Juruá in western Acre, Brazil, is a persistent malaria transmission hotspot partly due to fish farming development that was encouraged to improve local standards of living. Fish ponds can be productive breeding sites for Amazonian malaria vector species, including *Nyssorhynchus darlingi*, which, combined with high human density and mobility, add to the local malaria burden.This study reports entomological profile of immature and adult *Ny. darlingi* at three sites in Mâncio Lima, Acre, during the rainy and dry season (February to September, 2017). From 63 fishponds, 10,859 larvae were collected, including 5,512 first-instar Anophelinae larvae and 4,927 second, third and fourth-instars, of which 8.5% (n = 420) were *Ny. darlingi*. This species was most abundant in not-abandoned fishponds and in the presence of emerging aquatic vegetation. Seasonal analysis of immatures in urban landscapes found no significant difference in the numbers of *Ny. darlingi*, corresponding to equivalent population density during the rainy to dry transition period. However, in the rural landscape, significantly higher numbers of *Ny. darlingi* larvae were collected in August (IRR = 5.80, p = 0.037) and September (IRR = 6.62, p = 0.023) (dry season), compared to February (rainy season), suggesting important role of fishponds for vector population maintenance during the seasonal transition in this landscape type. Adult sampling detected mainly *Ny. darlingi* (~93%), with similar outdoor feeding behavior, but different abundance

**Data Availability Statement:** The data are available in the VectorBase repository (VBP0000323),

https://legacy.vectorbase.org/popbio/project/?id=
VBP0000323.

**Funding:** This study was partially funded by grants from the US National Institutes of Health ICEMR U19 AI089681 to Joseph M. Vinetz and Tropical Disease Research-WHO Contract 201460655 to Dionicia Gamboa.

**Competing interests:** The authors have declared that no competing interests exist.

according to landscape profile: urban site 1 showed higher peaks of human biting rate in May (46 bites/person/hour), than February (4) and September (15), while rural site 3 shows similar HBR during the same sampling period (22, 24 and 21, respectively). This study contributes to a better understanding of the larvae biology of the main malaria vector in the *Vale do Rio Juruá* region and, ultimately will support vector control efforts.

## Introduction

The link between anthropogenic environmental change and the emergence of malaria is well-documented in the Amazon basin [1–3]. Increased human population and land use/land cover change (LULC) influence the biological community, including Anophelinae mosquitoes, particularly those with some degree of synanthropy and competence to transmit *Plasmodium* sp. that circulate in the Amazon region [4]. This vast region is responsible for 99.5% of human malaria in Brazil, mainly *Plasmodium vivax* (>90% in 2019) [5]. Disease indicators vary according to the types of LULC and the socio-environmental aspects of occupied environments, influencing spatiotemporal malaria distribution trends [6]. Although from 2008–2016 Brazil reported annual reductions of the disease, with 2016 having the lowest incidence in the past 35 years, in 2017 the incidence increased by 50% compared with the previous year, decreasing only in 2019 [5]. This resurgence emphasizes the need for routine and integrated surveillance, even when disease rates are low, a characteristic feature of seasonal infectious diseases [7]. A key factor involved in the successful eradication policy of mosquito-borne diseases with a broad distribution and different focal transmission, such as malaria in the Amazon, is the identification and characterization of vector sources, following evaluation of potential tools for an integrated intervention framework [8].

Fish farming has been associated with malaria risk in the Amazon in Brazil [9], Peru [10], Colombia [11,12], and in sub-Saharan Africa in Nigeria [13] and Cote d'Ivoire [14]. The Vale do Juruá, Mâncio Lima municipality, is a classic example of the potential hazards of extensive fish farming in a periurban/urban setting. A local government program provided resources to residents to construct fish farms, frequently located in their backyards. The unwanted effect of this development program was the increased number of suitable larval habitats of *Nyssorhynchus darlingi* and other local malaria vectors which affect density and spatial distribution and threaten control strategies in the area [15–17]. Nowadays, the Vale do Juruá in western Acre is the region with the highest malaria numbers in Brazil, for both *P. vivax* and *P. falciparum*. In a scenario where anthropogenic fish farms have been demonstrated to be major contributors to vector abundance and *Plasmodium* transmission, larval source management (LSM) can be a practical component of integrated vector management (IVM) to reduce or eliminate immature stages of mosquito vectors [18–20]. Further, the recognition that variation in larval habitats, particularly in nutrient availability, strongly influences mosquito fitness, longevity, and malaria transmission dynamics, has renewed interest in larval environments [21,22]. On the other hand, LSM as part of a vector-borne disease control management plan has limitations when dealing with natural aquatic habitats in rural and forest areas, especially when breeding sites are extensive, inaccessible, and require frequent intervention such as clearing aquatic vegetation [23,24]. To address the application and effectiveness of any control strategies on mosquito borne-disease transmission, local vector biology information is essential, considering the diversity of *Ny. darlingi* in different environmental profiles of the Amazon Basin, reflected in malaria epidemiology. Although entomological surveys addressing

Anopheline larvae and the main vector *Ny. darlingi* presence in fishponds have been conducted in the Vale do Jurua [15–17], these studies did not focus on follow-up with short intervals between observations (one/two months per collection), nor characterize environmentally the fishponds associated with larvae sampled.

In the present study, an entomological survey of larvae and adult malaria vectors was conducted to evaluate the presence of the main vector *Ny. darlingi* in fishponds and neighboring households in Mâncio Lima, Acre. To address this, our study examined: (i) aquatic habitat parameters associated with Anophelinae larval abundance; (ii) differences in the abundance of *Ny. darlingi* during the rainy to dry seasonal transition; (iii) the microgeographic effect of urban and rural landscapes on the population dynamics of *Ny. darlingi*; and (iv) a comparison of human biting rates (HBR) and patterns of *Ny. darlingi* biting times influenced by different landscape scenarios.

## Methods

### Ethics statement

This study was approved by the World Health Organization Ethics Review Committee (0002669). Verbal consent was obtained from residents for collections on their properties, with the collaboration of the Mâncio Lima Endemics Diseases Coordination. A monthly report of fishpond physiochemical conditions was provided to each resident. Adult captures were conducted only by the authors, who used antimalarial prophylaxis as recommended by the Brazilian Ministry of Health.

### Study area

The municipality of Mâncio Lima is located in western Acre state, Brazil (7° 36' 50" S 72° 53' 45" W) along Highway BR 364 (Fig 1). An Anophelinae larval survey in artificial and natural breeding-sites reported four times more immatures in fishponds [15] compared with natural habitats. A time-series analysis (2003 to 2013), strongly suggested a spatiotemporal association between fish farming and malaria incidence [16]. The estimated population of Mâncio Lima is 17,545 [25], with the municipality registering for *P. vivax*: 6,632 infections in 2016 (API = 378 per 1000 habitants) and 7,049 infections in 2017 (API = 400); for *P. falciparum*: 1,172 in 2016 (API = 70) and 1,752 in 2017 (API = 99.8) (http://www2.datasus.gov.br/DATASUS, 2018). Notifications for monthly malaria shows significant linear correlation (>0.5) with rainfall: for *P. vivax* in 2016: $r = 0.75$, in 2017: $r = 0.43$; for *P. falciparum*: in 2016: $r = 0.51$, in 2017: $r = 0.47$ (S1 Fig). The most recent livestock census (2016) registered a total of 5,392 cattle in Mâncio Lima, mainly in rural areas (unpublished document, Institute of Agriculture and Forestry Defense of Acre, 2016).

### Study design

This research entailed an observational study of malaria vector ecology. For the Anophelinae survey, independent geographical areas were delimited based on two sampling criteria: the presence of a human residence occupied for at least the past 12 months for adult mosquito collection, and nearby fishponds for larval collection, whether economically active (used for pisciculture at the moment of the survey), or abandoned. Two perimeters (500 m and 1000 m) were virtually attributed for each residence to delimit each study site, and to support the localization of fishponds (Fig 1). These distances were chosen based on the flight range of *Ny. darlingi* in a rural settlement in Rondonia state, between 500 and 1000 m [26]. To test the influence of an urban area on local transmission, two sites (Sites 1 and 2) were selected near

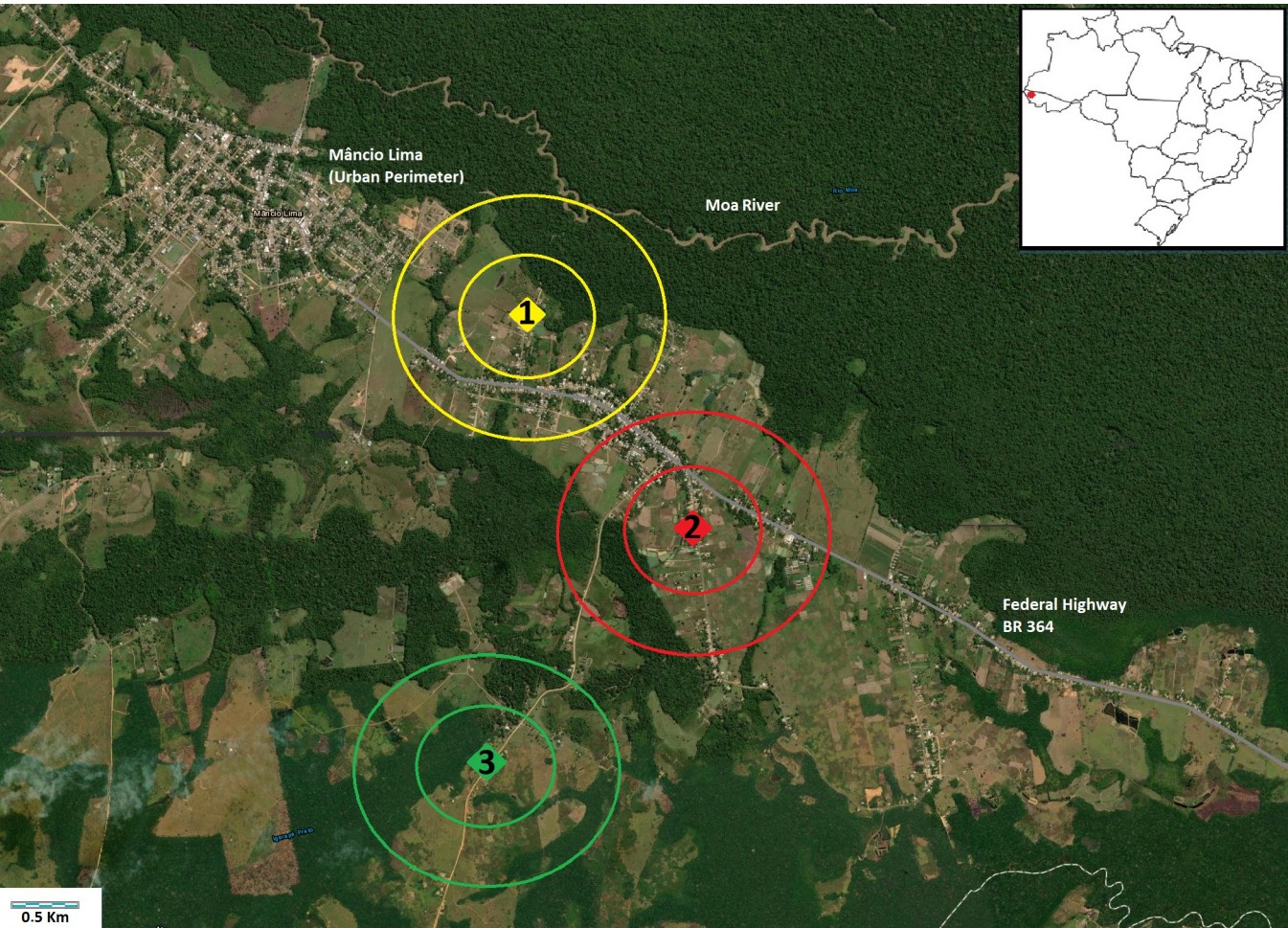

**Fig 1. Satellite image of Mâncio Lima municipality, showing the three study sites.** Site 1, urban, yellow, near Federal Highway BR 364 and Mâncio Lima town; Site 2, urban, red, near Federal Highway BR 364 and more distant from Mâncio Lima; Site 3, rural, green, distant from both BR 364 and Mâncio Lima. Each site shows the residence at center, and the two perimeters: 0.5 and 1.0 km. The insert is a map of Brazil indicating the location of Mancio Lima in Acre state. Content is the intellectual property of Esri and is used herein with permission. Copyright © 2021 Esri and its licensors. All rights reserved.

Federal Highway BR 364; and one site (Site 3) that was more distant from the highway (Fig 1). Highway BR 364 is important for socio-economic landscape concepts in Acre state: usually, urban landscape profiles include paved streets and have several residences and other human dwellings (schools, hospital, commercial facilities), and this may be reflected in a higher number of families and houses, leading to the establishment of more fishponds. On the other hand, rural landscape profiles consist of a lower human presence, fewer dwellings, and primary or secondary forest cover, if the landscape has not been exploited for logging, agriculture, or live-stock [27]. The presence of at least one fishpond near the house (within at least 500 m), positive previous larvae and adult captures (in December 2016), ease of access to the property, and co-operation of the residents were other considerations for the three residence selections and the respective representative sites.

## Larval and adult capture

Monthly larval collections were performed for six months in 2017 spanning rainy and dry sea-son (February, March, April, May, August, and September). Each fishpond was sampled by 1)

determining fixed sampling-points along fishpond margins (n = 4, A-D); and 2) sampling by dipper at 10 sampling-points along each margin. The 10 dips were evenly distributed according to the length of each margin. Dippers were standard: 10 cm in diameter, with a volume of 350 ml and a 1.5 m long handle, and white in color for better visibility of immatures [28]. Larval specimens were placed in 50 ml microtubes labeled according to sampling date, site, fishpond number and sampling-point margin letter (A-D) and number (1–10). All material was fixed in the field in 80% ethanol. Presence of aquatic fauna collected in the dippers were also recorded (i.e., *Culex* sp., amphibians, fish).

Adult collections were performed at each of the three sites in February, May, and September 2017. We used human landing catch (HLC), performed only by the professionally trained authors (two people indoors and two peridomestic simultaneously, rotating every two hours at each spot), using manual aspirators to capture mosquitoes, for 12 h /night (18:00–06:00). Collected mosquitoes were separated by date, location, and hour of capture. In months and sites with low mosquito density, we sampled one additional night (12h) and adjusted later for analysis. In February, there were two night collections at Site 1, and one in Sites 2 and 3, respectively. In May and September, two collections were done at Site 2, and one in Sites 1 and 3. Mosquitoes were stored in silica gel in microtubes (50 ml) identified with a code that included: month, site, date, and hour of collection. On rainy nights, adult captures were suspended and conducted on the following non-rainy night.

Field-collected specimens were identified at the Laboratory of Infectious Diseases of the Federal University of Acre (UFAC—campus Cruzeiro do Sul, Acre state) and at the School of Public Health of the University of São Paulo (USP—campus São Paulo, São Paulo state). Adults and the larval stages L2–L4 were identified using a stereomicroscope and entomological keys (Forattini, 2002). Because of the challenge to identify L1 morphologically [29], three larval groups were defined: Anophelinae L1 stage; Anophelinae L2–L4 stages and *Ny. darlingi* L2–L4 stages: in this approach, Anophelinae L2–L4 group included no *Ny. darlingi* species. After morphological identification, adults and larvae were sent to the Biotechnological Institute of University of State of São Paulo (UNESP—Campus Botucatu, São Paulo State) for further molecular analysis.

## Environmental variables

Fishponds were classified and measured according to environmental and physical-chemical conditions. For the environment, categorical variables included periodicity (permanent or temporary during the 6-month study period); abandoned fishpond- no maintenance by the owner (yes or no); associated vegetation on the margins of fishpond (if present: emerging, submerged, floating); the presence of *Culex* sp., amphibians, and fish. For periodicity and abandoned by the owner, classification was at the fishpond level; for vegetation and presence of other animals, classification was at the sampling-point level. Physical-chemical variables included pH, temperature, and conductivity, measured using an ExTECH multiparameter (extech.com/) probe that presented continuous values. However, due to functionality limitations, data from this device were collected only in the first three months (February, March, April). For the remaining three months (May, August, September), pH, nitrates (mg/L), nitrites (mg/L), carbonate hardness (KH) and dissolved chlorine (mg/L) were collected using a JBL ProScan kit (jbl.de/en), by immersion of a test strip in the water and reading by smartphone app downloaded at Google Play Store (play.google.com/store/apps/details?id=de.jbl.proscan). The data collected using the JBL Proscan kit had a more limited range, i.e., categorical variables. Collections using the ExTECH multiparameter probe were obtained at the sampling-point level; for the JBL Proscan test kit, data were obtained at the fishpond level.

Turbidity and shading were also obtained only during the last three months of the survey (May, August, September). Water turbidity was determined at the fishpond level using a LaMotte (lamotte.com) water column test kit, with discrete values ranging between 0–200 JTU (where 0 represents translucent water), at the fishpond level. Shading by canopy was collected at the sampling-point level with a TerraGes spherical densitometer according to the manufacturer's specifications (terrages.pt), with continuous values ranging between 0–24.96 1⁄4"-squares (where 0 represents shaded and 24.96 represents completely exposed), at the sampling-point level. This information is summarized in S1 Table.

Monthly precipitation data were obtained from the CPTEC/INPE website (clima1.cptec.inpe.br/). Adverse weather/air conditions (rain, mist, wind, smoke from burning) were noted when they occurred during the adult night collections. Field information was digitally stored through Open Data Kit (ODK). Data were compiled in EXCEL (Microsoft). Visual resources (photographs) were also obtained from each sampling-point, by ODK function. Georeferencing of the residences and fishponds was conducted using GPS Garmin device and Google Earth Pro TM software.

## Data setting and statistical analysis

Statistical analyses were conducted to establish the association between larval groups, environment, and physical-chemical variables, using multilevel regression models. The seasonal pattern, according to rainfall trends of western Amazon Basin (Rainy Season: Oct-Feb; Dry Season: Apr-Sept, see S1 Fig), was analyzed, considering the repeated measures framework used for larval sampling: the month of the collection was assumed to be a variable factor, with February being the chosen reference baseline according to rainfall seasonality effect on Culicidae biology abundance [29]. Therefore we chose February to represent a rainy month; September to represent a dry month; and the interval between February and September as the rainy-dry transition (S1 Fig). Larval counts of three groups (Anophelinae L1; Anophelinae L2 –L4; *Ny. darlingi* L2—L4) were considered the outcome variables.

Overdispersion was observed in data distribution resulting from large numbers of zero values, thus a binomial negative regression analysis was used [30]. According to assumptions of a negative binomial distribution, and the respective nature of dependent variables, regression coefficients are presented as incidence rate ratios (IRRs), defined by the number of events (Anophelinae counts) by fishpond (analysis unit) [31]. For all tests, the statistical significance level assumed was 0.05. An initial univariate regression was performed to verify any associations between single independent variables. Considering the non-randomized approach, multivariate regression was performed to verify adjustments in the coefficients. A cut-off value for *p* of less than 0.2 of univariate analysis was chosen, and the order of insertion of the independent variable in the multivariate regression was from the lowest to the highest *p*-value considering the univariate analysis [32]. Multicollinearity was assessed for the following independent variables used in multivariate analysis, since they were measured at the same sampling level: linear correlation for numerical variables (continuous physical-chemical) and Spearman rank correlation coefficient for ordinal variables (categorical physical-chemical).

Considering the hierarchical data structure (samplings-points nested within fishponds), a mixed-effects model was conducted, mainly due to its flexibility in repeated measures modeling of unbalanced data [33]. The dataset was structured in a long format, with the *i*th row functioning as a time-point per specific sampling-point, and respective fishpond (the subject of the analysis) [34]. Considering the biology of Anophelinae the three study sites were not considered independent, the usual procedure for mixed models that simulates repeated measures ANOVA, due to geographic proximity between sites (mainly Sites 1 and 2) [35]. In addition to

the overall regression, to distinguish effects among sites, regressions were performed for each site. An unconditional model was built first, followed by a model with a random component to indicate the subject of the repeated term. A two-level model was chosen, combining sampling-points at the first level and fishponds at the second level as the random component, according to the data structure (sampling-points nested in fishponds). Due to some gaps in variables measured during the monthly survey, a full dataset was the primary design (six-months), using the respective independent environmental variables: periodicity, abandoned, associated vegetation, presence of animals, and collection month. For physical-chemical variables, the turbidity of water and shading, which were not possible to measure during the whole six-month survey, three-month datasets were designed according to independent variables: pH, temperature and conductibility (continuous data) in February, March, and April; shading, turbidity and pH, nitrates, nitrites, carbonated hardness and dissolved chlorine (ordinal data), measured in May, August, and September. For three-month physical-chemical datasets, individual regressions were not performed for each site, due to reduced sampling effort. This information is summarized in S1 Table.

For comparison of categorical variables between sites, as well as the hypothesized adult abundance difference between indoor and outdoor, a Chi-square test was used. Outliers and systematic errors were verified through box-plot graphs. All statistical analyses were performed using Stata 14.2 (data analysis and statistical software—StataCorp LP, College Station, TX, USA). A robust option for the variance component estimators (VCE) was chosen according to the Stata configuration.

## Results

### Sampling sites

Sixty-three fishponds in the three sites were identified and followed during the 2017 study period. Total numbers of fishponds monitored throughout the field survey was variable because of seasonal precipitation or occasionally being emptied by owners, and some fishponds could not be reached across flooded fields at some point during the sampling period. Fig 2 shows satellite images for the three sites with each nearby residence and the fishponds surveyed; number of fishponds sampled by period, along with dry conditions and other characteristics is presented in S2 Table. S3 Table shows environment variables by site: both urban Sites 1 (~66%, 92/139) and 2 (~88%, 126/143) show a higher number of not abandoned fishponds compared with rural Site 3 (~22%, 12/64) ($p < 0.001$). No significant difference in fishpond periodicity was identified between three sites ($p = 0.625$); all sites had high numbers of permanent fishponds: (Site 1: ~86%, 120/139; Site 2: 85% 122/143; Site 3: ~84%, 46/55).

### Larval collection

During the six-month sampling period of 2017, 10,859 larvae were collected: n = 5,512 corresponded to the group of Anophelinae L1 stage species; n = 4,927 to the group of Anophelinae L2–L4 stage species; and n = 420 to the group of *Ny. darlingi* L2–L4 stages. Urban Site 1 shows the highest number of larvae (n = 6,065), followed by rural Site 3 (n = 3,017) and urban Site 2 (n = 1,777). Rural Site 3 had the highest density of larvae per fishpond (54.85), followed by urban Sites 1 (43.63) and 2 (12.46). For the *Ny. darlingi* L2–L4 group, urban Site 1 had the highest total number (249) and density per breeding site (1.80); urban Site 2 had 97 and 0.68, respectively; rural Site 3 had 74 and 1.32, respectively.

Anophelinae species and *Ny. darlingi* (both in L2–L4 stages) distributed by fishpond, site, and period are shown in Fig 3. The higher proportion of non-*Ny. darlingi* Anophelinae species compared with *Ny. darlingi* in practically all fishponds during the rainy season from February

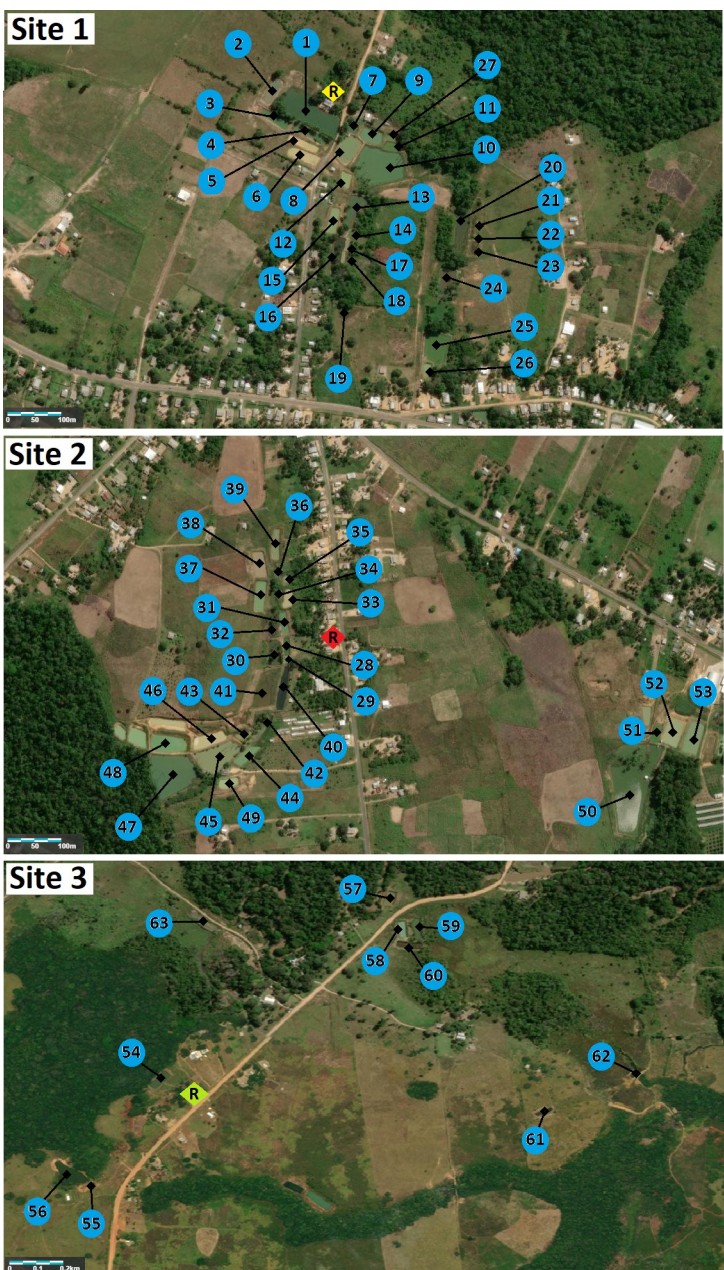

**Fig 2. Satellite image of three study sites, with respective residence (R) and fishponds.** Content is the intellectual property of Esri and is used herein with permission. Copyright © 2021 Esri and its licensors. All rights reserved.

through May (except for Fishpond number 07, Site 1, in February) is noteworthy. The exclusive presence of *Ny. darlingi* was observed in some fishponds at urban Site 2, however, these count values were minimal (1 or 2 specimens). For the dry season (August and September), urban Site 1 had more fishponds that were positive exclusively for *Ny. darlingi*: 01 in August; 01, 04, 08, 11 18, 25, and 26 in September. Urban Site 2 also had some fishponds with *Ny. darlingi* exclusively, but also with low counts. Rural Site 3 had *Ny. darlingi* in August and September in fishponds 54, 55 and 58 where this species was not observed during the rainy season months.

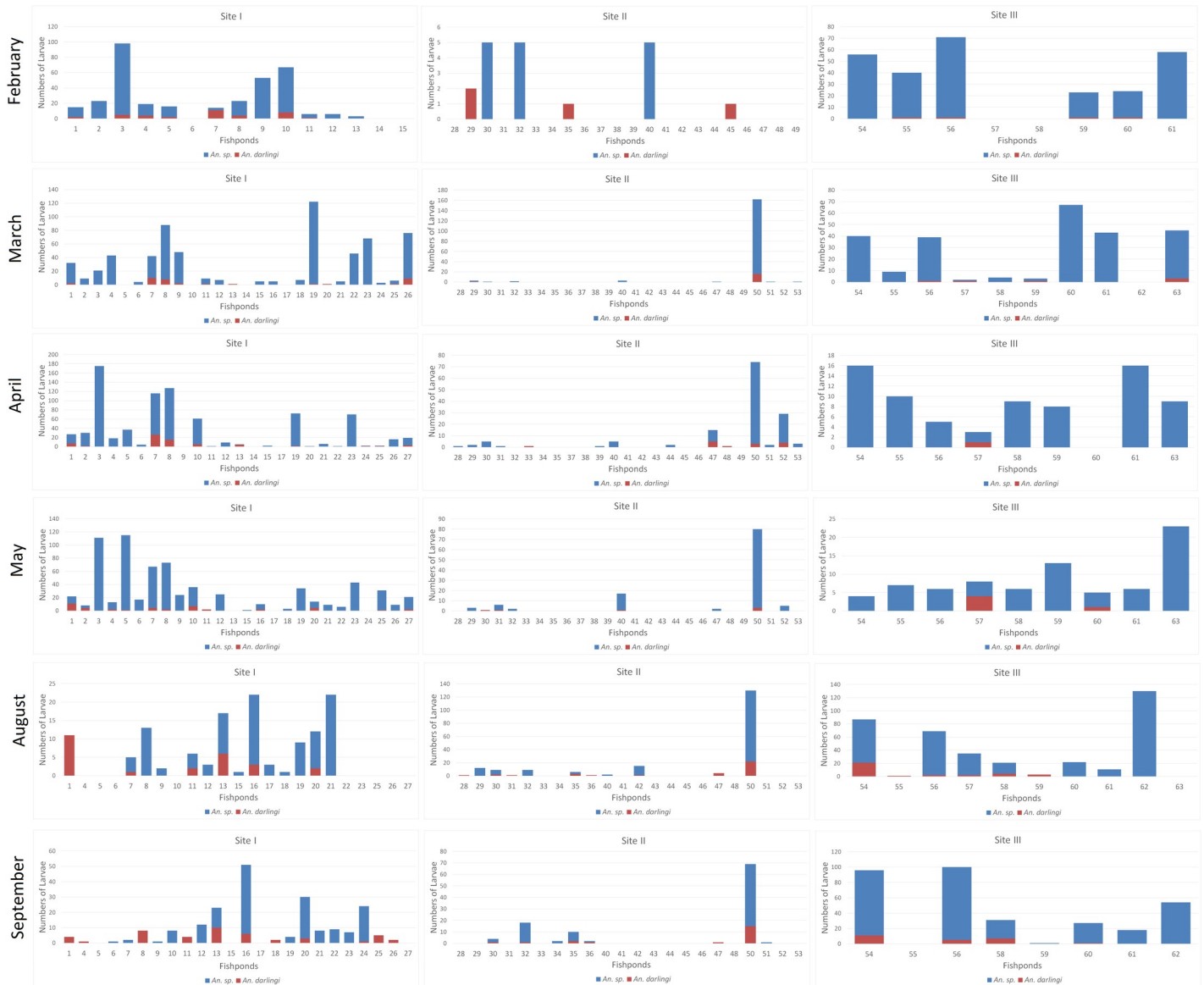

**Fig 3. Summary of larvae collected in the study.** Anophelinae spp. *and Ny. darlingi* (both L2 to L4 stages), distributed by fishpond, site and period.

### Statistical analysis

Table 1 shows the IRR coefficient results for the six-month dataset using univariate analysis, by identified Anophelinae larval group. Table 2 shows the multivariate analysis results, according to the selection criteria for independent variables. The Spearman rank coefficient detected no correlation between independent variables of the six-month dataset, with values lower than 0.1, except for the presence of *Culex* sp. and amphibians (0.34), and submerged aquatic vegetation with floating vegetation (0.23).

Considering the February baseline value and a statistical significance at 95% C.I., seasonality differences were not detected in the overall regression for the three identified Anophelinae larval groups. For each site, Anophelinae L1 shows a particular pattern in the univariate regression: a decrease in counts for August and September in urban Site 1, and April and May in

**Table 1. Incidence Rate Ratio (IRR), 95% Confidence Interval and *p* values for Anophelinae identity-group, for six month survey univariate two level negative binomial regression.**

| Anophelinae identity-group | Independent Variable | | Univariate Two-level Negative Binomial | | | | | | | |
|---|---|---|---|---|---|---|---|---|---|---|
| | | | Overall | | Site 1 | | Site 2 | | Site 3 | |
| | | | IRR (95% C.I.) | *p* | IRR (95% C.I.) | *p* | IRR (95% C.I.) | *p* | IRR (95% C.I.) | *p* |
| Anophelinae species (L1) | Month Collection | February | 1 | | 1 | | 1 | | 1 | |
| | | March | 1.23 (.78–1.92) | 0.373 | 1.19(.64–2.23) | 0.572 | .87 (.23–3.26) | 0.839 | 1.19(.61–2.30) | 0.613 |
| | | April | 1.29 (.73–2.25) | 0.374 | 1.16(.58–2.33) | 0.658 | 3.75(.90–15.60) | 0.069 | .39(.18 − .82) | 0.013 |
| | | May | .94 (.55–1.61) | 0.842 | 1.07(.56–2.06) | 0.822 | 1.05(.28–3.89) | 0.940 | .35(.13 − .81) | 0.016 |
| | | August | .56 (.24–1.28) | 0.175 | .13(.04 − .35) | 0.000 | 4.21(1.27 − 13.99) | 0.019 | .64 (.19–2.20) | 0.487 |
| | | September | .63 (.29–1.34) | 0.256 | .25(.09 − .64) | 0.004 | 1.99 (.63–6.29) | 0.239 | 1.21(.38–3.85) | 0.742 |
| | Periodicity | Temporary | 1 | | 1 | | 1 | | 1 | |
| | | Permanent | 2.19 (.55–8.70) | 0.262 | 1.32 (.45–3.87) | 0.608 | 5.01 (.73–34.07) | 0.100 | 2.12(1.20 − 3.75) | 0.009 |
| | Abandoned | Yes | 1 | | 1 | | 1 | | 1 | |
| | | No | .37 (.13–1.02) | 0.056 | 1.63 (.70–3.77) | 0.249 | 2.49 (.26–23.44) | 0.423 | 1.08 (.32–3.59) | 0.900 |
| | Associated Vegetation | Emerging | 1.86(1.07 − 3.25) | 0.021 | 1.05 (.43–2.59) | 0.906 | 3.98(1.25 − 12.68) | 0.020 | 1.45 (.93–2.28) | 0.097 |
| | | Submerged | .74 (.39–1.38) | 0.343 | .48 (.20–1.14) | 0.101 | .83 (.39–1.75) | 0.629 | .64 (.16–2.47) | 0.521 |
| | | Floating | 1.06 (.60–1.89) | 0.874 | .73 (.31–1.70) | 0.473 | 1.83 (.84–4.01) | 0.129 | .82 (.46–1.47) | 0.511 |
| | Presence | *Culex* sp. | 1.73(1.16 − 2.57) | 0.007 | 1.69(1.04 − 2.74) | 0.032 | 1.79 (.64–5.05) | 0.264 | 1.84 (.9–3.78) | 0.094 |
| | | Amphibian | 1.53 (.96–2.44) | 0.073 | 1.32 (.86–2.03) | 0.193 | 5.78 (.56–59.4) | 0.140 | 1.34 (.59–3.01) | 0.474 |
| | | Fish | 3.34(1.58 − 7.04) | 0.002 | 5.55(2.19 − 14.09) | 0.000 | .45 (.072–2.84) | 0.398 | 2.09 (.72–6.06) | 0.175 |
| Anophelinae species (L2, L3, L4) | Month Collection | February | 1 | | 1 | | 1 | | 1 | |
| | | March | 1.12 (.69–1.79) | 0.649 | 1.30(.677–2.51) | 0.426 | 1.12 (.44–2.86) | 0.807 | .77 (.38–1.54) | 0.457 |
| | | April | 1.17 (.69–2.01) | 0.550 | 1.20 (.66–2.18) | 0.543 | 4.65(1.11 − 19.46) | 0.035 | .30(.17 − .54) | 0.000 |
| | | May | 1.07 (.63–1.82) | 0.796 | 1.25 (.64–2.46) | 0.512 | 1.93 (.72–5.21) | 0.193 | .33(.15 − .73) | 0.007 |
| | | August | .85 (.38–1.92) | 0.709 | .28(.08 − .87) | 0.028 | 3.51(1.22 − 10.18) | 0.020 | 1.09 (.33–3.67) | 0.878 |
| | | September | .87 (.41–1.84) | 0.717 | .49 (.16–1.56) | 0.232 | 2.28 (.72–7.25) | 0.161 | 1.11 (.45–2.95) | 0.835 |
| | Periodicity | Temporary | 1 | | 1 | | 1 | | 1 | |
| | | Permanent | 2.88 (.54–15.31) | 0.214 | 1.24 (.21–7.33) | 0.805 | 13.9(1.54 − 125.35) | 0.019 | 1.69 (.82–3.5) | 0.150 |
| | Abandoned | Yes | 1 | | 1 | | 1 | | 1 | |
| | | No | .44 (.15–1.26) | 0.129 | 2.34 (.83–6.64) | 0.107 | 1.82 (.18–18.42) | 0.610 | 1.15 (.35–3.77) | 0.811 |
| | Associated Vegetation | Emerging | 1.82(1.01 − 3.31) | 0.050 | .95 (.43–2.07) | 0.902 | 3.71(1.06 − 13.04) | 0.040 | 1.89 (.68–5.19) | 0.217 |
| | | Submerged | .55 (.27–1.12) | 0.098 | .29(.13 − .68) | 0.004 | .56 (.22–1.45) | 0.235 | .98 (.21–4.60) | 0.985 |
| | | Floating | 1.41 (.78–2.53) | 0.252 | 1.37 (.61–3.10) | 0.445 | 1.77 (.65–4.86) | 0.264 | .73(.64 − .83) | 0.000 |
| | Presence | *Culex* sp. | 1.78(1.16 − 2.74) | 0.009 | 1.86(1.09 − 3.19) | 0.023 | 1.06 (.32–3.57) | 0.917 | 2.2(1.1 − 4.39) | 0.025 |
| | | Amphibian | 1.37 (.74–2.52) | 0.313 | 1.11 (.55–2.23) | 0.762 | 5.44 (.87–33.83) | 0.069 | 1.29 (.46–3.65) | 0.620 |
| | | Fish | 2.92(1.23 − 6.93) | 0.015 | 2.84(1.08 − 7.51) | 0.035 | 3.92 (.33–46.71) | 0.279 | 4.18 (.54–32.11) | 0.169 |
| *Ny. darling* (L2, L3, L4) | Month Collection | February | 1 | | 1 | | 1 | | 1 | |
| | | March | .74 (.34–1.60) | 0.449 | .55 (.18–1.65) | 0.294 | .94 (.22–3.93) | 0.928 | 1.31 (.35–4.93) | 0.683 |
| | | April | 1.01 (.48–2.11) | 0.984 | .89 (.32–2.45) | 0.831 | 1.82 (.27–12.35) | 0.538 | .21 (.02–2.26) | 0.194 |
| | | May | .81 (.41–1.57) | 0.532 | .74 (.30–1.81) | 0.512 | .56 (.11–2.79) | 0.480 | 1.03 (.14–7.58) | 0.971 |
| | | August | 1.39 (.57–3.41) | 0.465 | .39 (.09–1.58) | 0.190 | 3.01 (.75–11.94) | 0.118 | 5.80(1.08 − 30.94) | 0.039 |
| | | September | 1.55 (.68–3.54) | 0.291 | .91 (.28–2.88) | 0.879 | 1.61 (.42–6.09) | 0.481 | 6.25(1.19 − 32.76) | 0.030 |
| | Periodicity | Temporary | 1 | | 1 | | 1 | | 1 | |
| | | Permanent | 1.21 (.38–3.77) | 0.742 | .86 (.22–3.36) | 0.835 | 5.89 (.46–75.12) | 0.172 | .654 (.18–2.27) | 0.504 |
| | Abandoned | Yes | 1 | | 1 | | 1 | | 1 | |
| | | No | 1.81 (.66–4.95) | 0.243 | 10.27(2.7 − 39.07) | 0.001 | 1.00 (.21–4.87) | 0.996 | .91 (.19–4.27) | 0.906 |
| | Associated Vegetation | Emerging | 2.08(1.25 − 3.46) | 0.005 | 1.61 (.94–2.76) | 0.079 | 3.07 (.96–9.78) | 0.058 | 2.08 (.49–8.72) | 0.318 |
| | | Submerged | 1.45 (.68–3.07) | 0.334 | 1.27 (.50–3.22) | 0.611 | .79 (.26–2.42) | 0.692 | 2.77 (.43–17.63) | 0.279 |
| | | Floating | 1.37 (.73–2.57) | 0.320 | 1.62 (.73–3.60) | 0.232 | 1.48 (.43–5.0) | 0.527 | * | 0.000 |
| | Presence | *Culex* sp. | 1.23 (.73–2.07) | 0.433 | 1.27 (.72–2.25) | 0.407 | 2.2 (.55–8.78) | 0.261 | .57 (.21–1.55) | 0.274 |
| | | Amphibian | 1.32 (.63–2.76) | 0.455 | 1.05 (.53–2.08) | 0.888 | 6.53(2.34 − 18.21) | 0.000 | .88 (.065–11.78) | 0.924 |
| | | Fish | 1.24 (.6–2.57) | 0.558 | 1.55 (.64–3.73) | 0.328 | .71 (.11–4.85) | 0.723 | 1.48 (.22–9.81) | 0.683 |

Statistically significant values at the 0.05 level are highlighted.

*IRR value omitted due low decimal number ($10^{-10}$).

**Table 2. Incidence Rate Ratio (IRR), 95% Confidence Interval and *p* values for Anophelinae species identity-group, for six-month survey multivariate two level negative binomial regression.**

| Anophelinae identity-group | Independent Variable | | Multivariate Two-level Negative Binomial | | | | | |
|---|---|---|---|---|---|---|---|---|
| | | | Site 1 | | Site 2 | | Site 3 | |
| | | | IRR (95%C.I.) | *p* | IRR (95%C.I.) | *p* | IRR (95%C.I.) | *p* |
| Anophelinae species (L1) | Month Collection | February | 1 | | 1 | | 1 | |
| | | March | 1.14 (.56–2.31) | 0.721 | .89 (.23–3.50) | 0.870 | 1.21(.62–2.37) | 0.579 |
| | | April | 1.30 (.52–3.24) | 0.568 | 2.30(.57–9.28) | 0.242 | .33(.16 − .69) | 0.004 |
| | | May | 1.31 (.56–3.04) | 0.534 | 1.07 (.29–3.92) | 0.916 | .25(.13 − .50) | 0.000 |
| | | August | .21(.07 − .56) | 0.003 | 3.70(1.05 − 13.03) | 0.041 | .54 (.16–1.79) | 0.317 |
| | | September | .31(.10 − .91) | 0.033 | 1.19 (.33–4.22) | 0.789 | 1.19(.31–4.55) | 0.797 |
| | Periodicity | Temporary | | | 1 | | 1 | |
| | | Permanent | | | 2.98 (.53–16.77) | 0.214 | 1.98 (.88–4.45 | 0.096 |
| | Associated Vegetation | Emerging | | | 5.27 (.91–30.46) | 0.063 | 2.79(1.51 − 5.17) | 0.001 |
| | | Submerged | .67 (.38–1.20) | 0.182 | | | | |
| | | Floating | | | .62 (.14–2.73) | 0.531 | | |
| | Presence | *Culex* sp. | 1.50 (.99–2.27) | 0.056 | | | 1.07 (.66–1.74) | 0.773 |
| | | Amphibian | .97 (.55–1.73) | 0.932 | 6.10(1.13 − 32.94) | 0.036 | | |
| | | Fish | 1.94 (.72–5.23) | 0.192 | | | 2.03 (.76–5.39) | 0.157 |
| Anophelinae species (L2, L3, L4) | Month Collection | February | 1 | | 1 | | 1 | |
| | | March | 1.43 (.76–2.67) | 0.268 | 1.09 (.42–2.80) | 0.864 | .81 (.37–1.76) | 0.593 |
| | | April | 2.00(1.11 − 3.59) | 0.020 | 2.56 (.73–8.99) | 0.141 | .46 (.17–1.25) | 0.129 |
| | | May | 2.48(1.29 − 4.76) | 0.006 | 1.64 (.58–4.62) | 0.351 | .46 (.15–1.41) | 0.175 |
| | | August | .55 (.18–1.68) | 0.293 | 2.74 (.95–7.88) | 0.062 | 1.71 (.55–5.33) | 0.353 |
| | | September | .81 (.26–2.50) | 0.721 | 1.46 (.31–6.77) | 0.628 | 2.40 (.60–9.58) | 0.214 |
| | Periodicity | Temporary | | | 1 | | 1 | |
| | | Permanent | | | 9.01(1.36 − 59.77) | 0.023 | 1.12 (.61–2.06) | 0.721 |
| | Abandoned | Yes | 1 | | | | | |
| | | No | 1.65 (.76–3.56) | 0.201 | | | | |
| | Associated Vegetation | Emerging | | | 3.04(1.14 − 8.15) | 0.027 | 66 (.22–1.96) | 0.454 |
| | | Submerged | .34(.19 − .58) | 0.000 | | | | |
| | Presence | *Culex* sp. | 2.24(1.36 − 3.69) | 0.001 | | | 1.96 (.82–4.72) | 0.131 |
| | | Amphibian | | | 6.47(1.89 − 22.14) | 0.003 | | |
| | | Fish | 1.12 (.43–2.86) | 0.819 | | | 6.14 (.86–43.76) | 0.070 |
| *Ny. darlingi* (L2, L3, L4) | Month Collection | February | 1 | | 1 | | 1 | |
| | | March | .65 (.21–1.98) | 0.452 | 1.11 (.22–5.54) | 0.897 | 1.31 (.35–4.86) | 0.681 |
| | | April | 1.01 (.36–2.81) | 0.989 | 1.36 (.20–9.14) | 0.753 | .21 (.02–2.31) | 0.204 |
| | | May | .66 (.26–1.64) | 0.371 | .83 (.16–4.36) | 0.825 | 1.04 (.14–7.55) | 0.967 |
| | | August | .48 (.11–2.00) | 0.314 | 2.71 (.57–12.88) | 0.209 | 5.80(1.11 − 30.41) | 0.037 |
| | | September | 1.11 (.34–3.65) | 0.864 | 1.32 (.31–5.67) | 0.705 | 6.62(1.29 − 33.89) | 0.023 |
| | Periodicity | Temporary | | | 1 | | | |
| | | Permanent | | | 3.84 (.55–26.57) | 0.173 | | |
| | Abandoned | Yes | 1 | | | | | |
| | | No | 11.40(3.06 − 42.52) | 0.000 | | | | |
| | Associated Vegetation | Emerging | 2.12 (.83–5.45) | 0.117 | 2.23 (.79–6.23) | 0.126 | | |
| | | Submerged | | | | | * | 0.000 |
| | Presence | Amphibian | | | 5.86(2.42 − 14.156) | 0.000 | | |

Statistically significant values at the 0.05 level are highlighted.

*IRR value omitted due low decimal number ($10^{-10}$).

rural Site 3; and an increase in counts for urban Site 2 in August. The IRR values observed in the multivariate analysis were maintained relative to the univariate IRR values, indicating that these results were not influenced by possible confounding factors. The Anophelinae L2—L4 group also shows a unique pattern for each site, however, in this case the IRR values were substantially different between the univariate and multivariate analysis. For the *Ny. darlingi* L2—L4 group, there was a similar pattern in both urban sites (1 and 2), with no significant statistical difference in monthly larval numbers for the baseline value (February). In rural Site 3, an increase was observed for August [5.8 (95% C.I.:1.11–30.41)] and for September [6.62 (95% C.I.:1.29–33.89)]. Multivariate and univariate regression showed comparable IRR values in the three sites. For periodicity and abandoned characteristics, permanent condition was significant for Anophelinae L1 group in rural Site 3, for univariate analysis only for the Anophelinae L2—L4 group, whereas urban Site 2 shows an increase in larval number for both univariate and multivariate regression. The non-abandoned condition for *Ny. darlingi* shows an increase in larval number in urban Site 1, for both univariate and multivariate regression. Emerging associated vegetation shows an increase of larval number in the overall regression for all three Anophelinae groups. Presence of *Culex* sp. and fish was significant for Anophelinae L1 and L2, L3 and L4, in the overall regression. Urban Site 1 showed a similar association for both groups, however only the Anophelinae L2—L4 group maintains this value in multivariate analysis. The presence of amphibians was positively associated with Anophelinae L1 and L2- L4 groups in urban Site 2 only in the multivariate regression. The *Ny. darlingi* group showed a positive association with amphibian presence only in urban Site 2, for both univariate and multivariate regressions.

Table 3 shows IRR coefficient results for a three-month dataset using univariate analysis, by identified Anophelinae larval group. Table 4 shows multivariate analysis, according to the selection criteria for independent variables. The Spearman rank test shows a high correlation between the categorical physical-chemical variables nitrates and nitrites ($r = 0.89$), and a low correlation between carbonate hardness and pH ($r = 0.38$), and carbonate hardness and dissolved chlorine ($r = 0.4$). Physical-chemistry variables for continuous values (pH, temperature, and conductibility) were not statistically associated at 95% C.I. with the abundance of any of the three larval groups. Turbidity shows a significant negative association for the Anophelinae L1 only in the univariate regression [0.98 (IC95%:.97-.99)]. *Ny. darlingi* L2—L4 shows a significant positive association with turbidity in multivariate analysis only [1.01 (IC95%:1.00–1.01), $p = 0.045$]. Shading reduction shows a significant negative association with the abundance of both Anophelinae L1, and Anophelinae L2—L4 in both univariate and multivariate regressions, but for the *Ny. darlingi* group, the univariate was not significant at the 0.05 level, however, it was near the limit [0.96 (IC95%:.93–1.00) with $p = 0.052$], whereas in the multivariate analysis shading was significant [0.95 (IC95%:.92-.99), $p = 0.02$].

For ordinal physical-chemistry variables, increased pH values were associated with decreased in larval counts in all three groups, for both univariate and multivariate regression. Similarly, the highest nitrate level (40 mg/L) was associated with decreased larval counts for all three larval groups, and this was maintained in multivariate analysis for the *Ny. darlingi* group. Nitrites were not significantly associated with larval counts, and excluded from the multivariate regression analyses. For carbonated hardness, whereas both Anophelinae L1 and L2, L3 and L4 groups show a highly significant negative association (except Anophelinae L1 at 15 KH in univariate analysis, not kept in the multivariate regression), the *Ny. darlingi* group did not show statistical significance for any range, except a decrease in larval numbers observed at a range of 3 KH in the multivariate analysis [0.24 (IC95%:.09-.63)]. Dissolved chlorine showed a significant positive association at a range of 1.5 mg/L for the Anophelinae L1 in multivariate

**Table 3. Incidence Rate Ratio (IRR), 95% Confidence Interval and p values for Anophelinae species identity-group, for three-month survey univariate two level negative binomial regression.**

| Independent Variable | | | Univariate Two-level Negative Binomial | | | | | |
|---|---|---|---|---|---|---|---|---|
| | | | Anophelinae species (L1) | | Anophelinae species (L2, L3, L4) | | *Ny. darlingi* (L2, L3, L4) | |
| | | | IRR (95% C.I.) | *p* | IRR (95% C.I.) | *p* | IRR (95% C.I.) | *p* |
| Physical-Chemistry (continual values) | pH | | 1.02 (.78–1.33) | 0.901 | .91 (.69–1.21) | 0.521 | 1.05(.72–1.54) | 0.779 |
| | Temperature | | .92 (.81–1.06) | 0.259 | .95 (.82–1.1) | 0.505 | .87 (.71–1.06) | 0.165 |
| | Conductibility | | 1.00 (.99–1.01) | 0.432 | 1.01 (.99–1.07) | 0.297 | 1.02 (.99–1.009) | 0.488 |
| Turbidity (discrete value) | | | .98(.97 − .99) | 0.050 | .98 (.97–1.00) | 0.120 | 1.01 (.99–1.02) | 0.100 |
| Shading (continual value) | | | .95(.91 − .99) | 0.008 | .96(.94 − .99) | 0.035 | .96 (.93–1.00) | 0.052 |
| Physical-Chemistry (categorical values) | pH | >6 | 1 | | 1 | | 1 | |
| | | 6.4 | .14(.05 − .39) | 0.000 | .06(.03 − .12) | 0.000 | .87 (.43–1.76) | 0.695 |
| | | 6.6 | .26 (.03–2.36) | 0.231 | .19 (.01–3.02) | 0.237 | 1.97 (.61–6.5) | 0.261 |
| | | 6.8 | .79 (.20–3.15) | 0.747 | .14 (.01–3.06) | 0.210 | .87 (.13–5.79) | 0.884 |
| | | 7 | .31 (.04–2.20) | 0.245 | .10(.02 − .46) | 0.003 | * | 0.000 |
| | | 7.2 | * | 0.000 | * | 0.000 | * | 0.000 |
| | | 7.6 | * | 0.000 | * | 0.000 | * | 0.00000 |
| | nitrates (mg/L) | 0 | 1 | | 1 | | 1 | |
| | | 10 | 1.53 (.51–4.64) | 0.449 | 1.39 (.51–3.83) | 0.519 | .91 (.38–2.19) | 0.839 |
| | | 25 | 1.39 (.22–7.84) | 0.705 | 1.36 (.23–7.95) | 0.734 | .72 (.15–3.59) | 0.688 |
| | | 40 | .22(.16 − .32) | 0.000 | .06(.04 − .09) | 0.000 | * | 0.000 |
| | nitrites (mg/L) | 0 | 1 | | 1 | | 1 | |
| | | 0.25 | 1.05 (.44–2.49) | 0.917 | .84 (.40–1.77) | 0.651 | .96 (.42–2.21) | 0.934 |
| | | 0.5 | 1.54 (.16–14.64) | 0.707 | 1.27 (.09–16.31) | 0.854 | .47 (.04–5.57) | 0.547 |
| | carbonated hardness (KH) | 0 | 1 | | 1 | | 1 | |
| | | 1.5 | 1.90 (.98–3.68) | 0.056 | 2.05(1.01 − 4.19) | 0.049 | .89 (.60–1.34) | 0.604 |
| | | 3 | .71 (.25–1.99) | 0.515 | 93 (.22–3.96) | 0.921 | .24 (.04–1.28) | 0.094 |
| | | 4.5 | .17(.06 − .47) | 0.001 | * | 0.000 | .30 (.08–1.2) | 0.089 |
| | | 6 | .10(.04 − .27) | 0.000 | * | 0.000 | .56 (.14–2.26) | 0.415 |
| | | 8 | .13(.06 − .26) | 0.000 | .33(.14 − .79) | 0.013 | .82 (.19–3.49) | 0.791 |
| | | 15 | 3.49(1.69 − 7.17) | 0.001 | .23(.12 − .43) | 0.000 | 2.08(.93–4.62) | 0.073 |
| | dissolved chlorine (mg/L) | 0 | 1 | | 1 | | 1 | |
| | | 0.8 | .94 (.26–3.39) | 0.922 | .66 (.15–2.99) | 0.588 | .82 (.27–2.51) | 0.730 |
| | | 1.5 | 2.04 (.94–4.42) | 0.070 | .72 (.28–1.88) | 0.506 | 2.21(1.17 − 4.19) | 0.015 |

Statistically significant values at the 0.05 level are highlighted.

*IRR value omitted due low decimal number ($10^{-10}$).

analysis [4.23 (IC95%:1.58–11.36)], and the *Ny. darlingi* for both univariate [2.21 (IC95%:1.17–4.19)] and multivariate analysis [3.41 (IC95%:1.51–7.68)].

## Adult collection

A total of 692 Anophelinae specimens was collected and identified as *Ny. darlingi*. Fig 4 shows HBR for each site, adjusted for two night captures depending on site (Site 1 in February; Site 2 in May and September). There was a significant difference in the proportion of indoor vs. outdoor *Ny. darlingi* among the 3 sites ($X^2$ = 19.833, $p<0.001$), with a higher abundance in the peridomestic area. The proportion indoors was higher in Site 3 (~25%) than in Site 1 (~12%)

**Table 4. Incidence Rate Ratio (IRR), 95% Confidence Interval and *p* values for Anophelinae species identify-group, for three-month survey multivariate two level negative binomial regression.**

| Independent Variable | | | Multivariate Two-level Negative Binomial | | | | | |
|---|---|---|---|---|---|---|---|---|
| | | | Anophelinae species (L1) | | Anophelinae species (L2, L3, L4) | | *Ny. darlingi* (L2, L3, L4) | |
| | | | IRR (95% C.I.) | *p* | IRR (95% C.I.) | *p* | IRR (95% C.I.) | *p* |
| Turbidity (discrete value) | | | .99 (.98–1.00) | 0.087 | .99 (.98–1.01) | 0.216 | 1.01(1.00 − 1.01) | 0.045 |
| Shading (continual value) | | | .95(.92 − .98) | 0.003 | .97(.94 − .99) | 0.041 | .95(.92 − .99) | 0.020 |
| Physical-Chemistry (categorical values) | pH | >6 | 1 | | 1 | | 1 | |
| | | 6.4 | .18 (.03–1.06) | 0.059 | .08(.025 − .24) | 0.000 | .28 (.06–1.37) | 0.118 |
| | | 6.6 | .71 (.06–8.19) | 0.782 | .68 (.04–10.58) | 0.781 | .84 (.12–5.62) | 0.856 |
| | | 6.8 | 6.95 (.35–136.97) | 0.202 | 4.73 (.49–45.68) | 0.179 | .53 (.04–7.78) | 0.645 |
| | | 7 | .34 (.02–6.38) | 0.473 | .27 (.03–2.73) | 0.268 | * | 0.000 |
| | | 7.2 | * | 0.000 | * | 0.000 | * | 0.000 |
| | | 7.6 | * | 0.000 | * | 0.000 | * | 0.000 |
| | nitrates (mg/L) | 0 | 1 | | 1 | | 1 | |
| | | 10 | 1.13 (.43–2.93) | 0.804 | 1.12 (.46–2.72) | 0.796 | .74 (.27–2.07) | 0.574 |
| | | 25 | .79 (.06–9.86) | 0.861 | .91 (.15–5.43) | 0.915 | .74 (.08–7.16) | 0.794 |
| | | 40 | .68 (.18–2.62) | 0.577 | .30 (.09–1.04) | 0.057 | * | 0.000 |
| | carbonated hardness (KH) | 0 | 1 | | 1 | | 1 | |
| | | 1.5 | 1.39 (.75–2.58) | 0.287 | 1.38 (.78–2.44) | 0.272 | .85 (.57–1.26) | 0.423 |
| | | 3 | .33(.14 − .75) | 0.008 | .39 (.18–1.21) | 0.103 | .24 (.09-.63) | 0.003 |
| | | 4.5 | .01(.01 − .52) | 0.020 | * | 0.000 | .13 (.01–1.88) | 0.135 |
| | | 6 | .03(.02 − .38) | 0.006 | * | 0.000 | .37 (.08–1.71) | 0.203 |
| | | 8 | .15(.05 − .46) | 0.001 | .25(.07 − .86) | 0.027 | 3.87 (.99–15.17) | 0.052 |
| | | 15 | .04(.02 − .51) | 0.013 | * | 0.000 | .34 (.03–4.23) | 0.401 |
| | dissolved chlorine (mg/L) | 0 | 1 | | | | 1 | |
| | | 0.8 | 1.67 (.63–4.37) | 0.299 | | | 1.34 (.64–2.78) | 0.435 |
| | | 1.5 | 4.23(1.58 − 11.36) | 0.004 | | | 3.41(1.51 − 7.68) | 0.003 |

Statistically significant values at the 0.05 level are highlighted.

*IRR value omitted due low decimal number ($10^{-10}$).

or Site 2 (~11%). In Site 1, May showed a higher number of *Ny. darlingi* in all night captures (21/173 indoor/outdoor) than February (3/13 indoor/outdoor) or September (5/29 indoor/outdoor). Site 2 showed the lowest adult collections: February (3/10 indoor/outdoor), May (0/8 indoor/outdoor), September (0/7 indoor/outdoor). In Site 3, mosquito numbers were consistently high for outdoor collections, and increased for indoor captures in the last two months: February (11/104, indoor/outdoor), May (51/101 indoor/outdoor), September (42/97, indoor/outdoor).

Regarding HBR per hour, Site 1 shows more activity in May between 19:00–20:00 (HBR = 46), while February presents low numbers between 19:00–20:00 (HBR = 4), as does September (18:00–19:00 = 13). Site 2 also presents low numbers (with peaks reaching at maximum of four mosquitoes/hr). In contrast, *Ny. darlingi* from Site 3 showed higher outdoors peaks during the first part of the night (February, 19:00–20:00 = 22; May, 18:00–19:00 = 24; September, 20:00–21:00 = 21), also demonstrating, besides low values, some indoor peaks that exceeded outdoors ones, in May (00:00–01:00, indoor = 12, outdoor = 7; 02:00–03:00, indoor = 7, outdoor = 0) and September (for 00:00–01:00, indoor = 10, outdoor = 7).

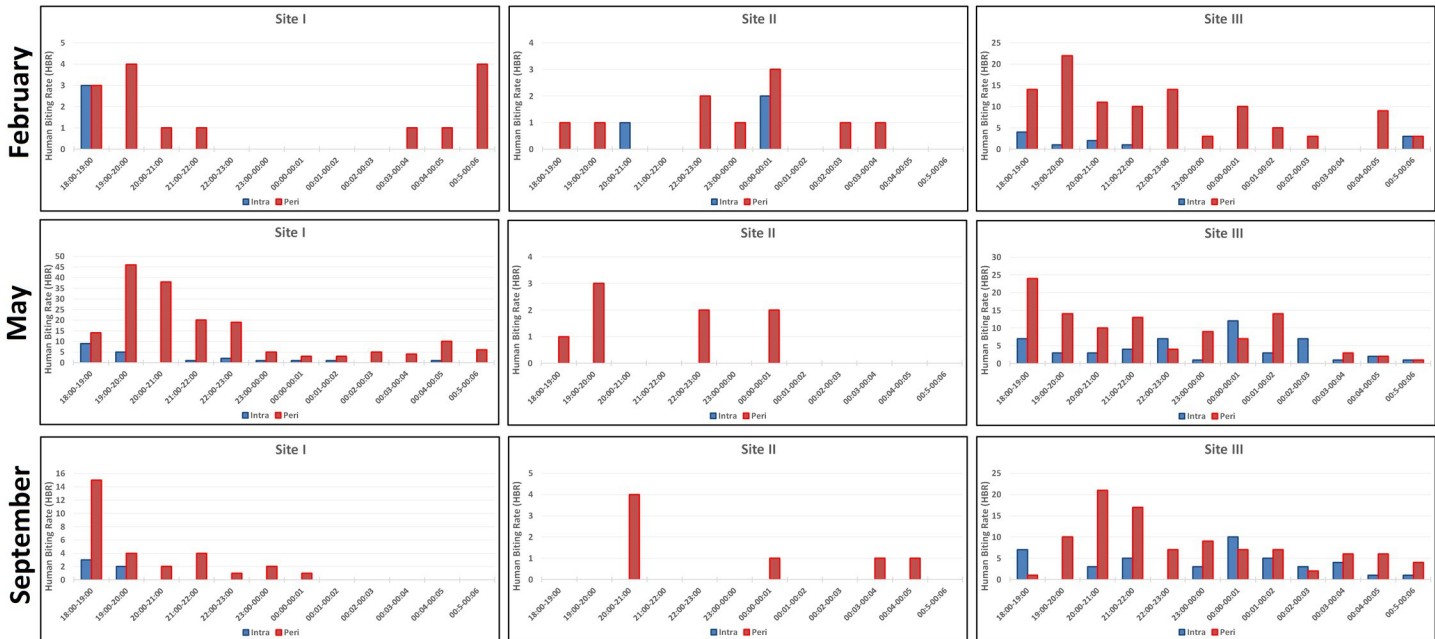

**Fig 4. Blood-feeding pattern by human biting rate (HBR: *Ny. darlingi* per human captures per hour), by night-capture, site and period.**

## Discussion

For effective control of Amazon malaria transmission it is essential to recognize the diverse eco-epidemiologic profiles of the disease in local areas: municipalities, cities, districts, subdistricts, along with "off the grid" areas: mining, rubber extraction (*seringal*), rural settlements and indigenous populations. For such heterogeneity, the design and the application of specific control methodologies according to each eco-epidemiologic profile is needed [36]. The Brazilian Amazonian Basin has a total area of five million km$^2$ (corresponding to an estimated 60% of the federal territory), but only 15% of the human population, most in big cities and state capitals [25]. This heterogeneous distribution is reflected in local characteristics of vector biology, thus malaria epidemiology, following human dynamics that drive Amazonian occupation [37]. Interdisciplinary methods for disease intervention are common but rarely tailored to specific local conditions [38,39]. For effective eradication at a global scale, many aspects of public health need to be included, such epidemiological and syndromic surveillance, early diagnosis, clinical treatment, environmental sanitation, and improved methods for economical land use to reduce inequity and poverty [40].

Ours is the first study to conduct a detailed microgeographic spatiotemporal analysis of larvae and adult Anophelinae, with a focus on the major vector, *Ny. darlingi*, in Vale do Jurua, western Acre, characterized by high malaria transmission associated with urban and periurban fishponds. In this area, we determined that *Ny. darlingi* larval dynamics was not affected by seasonality in urban landscapes, similar to findings in previous studies in the area [15,16]. This feature may help to maintain the population density of *Ny. darlingi* during the transition of rainy to dry seasons. We detected other fishpond characteristics associated with *Ny. darlingi* abundance: active fishponds, emergent vegetation (normally secondary growth that has emerged from deforested areas), and shade. A particular fishpond characteristic verified by the present study was the presence of *Ny. darlingi* larvae in water with dissolved chlorine, suggesting possible resilience for chemical pollution [41], although the increase of pH and nitrates

was observed as a limiting factor. Adult collections were conducted for a single night per study site per month and therefore our conclusions are preliminary. Most *Ny. darlingi* were collected outdoors, during the first part of the night (18:00–00:00), a pattern reported for this species in other Amazonian occupied areas [42–46], although we observed a greater abundance (not significant) and peaks of indoor activity in the rural landscape.

In Amazonian malaria transmission, the most common type of breeding site, whether natural or artificial (or both), contributes substantially to the dynamics and seasonality of malaria [26,47,48]. Two main variables of natural aquatic habitats that affect larval survival are water flow intensity during the rainy season (larval mortality rate); and low water capacity during the dry season (loss of available aquatic niche) [24,48]. These conditions are generally neutralized in artificial aquatic habitats such as dams, micro dams, cisterns, fishponds, and other types of flow-limited water bodies [23], increasing the vectorial capacity of primary vectors, such as *Ny. darlingi* [49]. A successful breeding site in a malaria-endemic region should provide geographic and temporal coexistence for the epidemiologic triad: vector, etiological agent, and human reservoir, according to ecological strategies of Anopheline species [50,51] as well *Plasmodium* sp. [52], facilitating adaptation to host behavior [53]. The presence of the primary malaria vector in human residences and adjacent fishponds in Mancio Lima suggests that transmission may occur both in and around houses, although our HBR data demonstrate that most biting occurs outdoors.

A lower proportion of *Ny. darlingi* larval specimens was identified in the present study (8.5%) compared to that found in the same municipality in earlier studies (16.1% [15]; 22.5% [54]), and in other distinctive local Amazonian environments [24,48]. However, *Ny. darlingi* L1 larvae were not morphologically identified herein and this stage represented more than 50% of the total numbers of larvae surveyed. Furthermore, similar to other entomological studies in malaria-endemic areas, our adult survey detected only *Ny. darling* [43–46], although we recognize that HLC can generate a bias due to the mainly anthropophilic behavior of this species, as well for *Nyssorhynchus* sp. in general. Biodiversity of Anophelinae can be an indicator of environmental disruption, a putative signal of future outbreaks [55]. There is both a notably increased abundance and/or the emergence of *Nyssorhynchus* species in human-colonized Amazonian areas [44,56], and low natural abundance of this genus in primary Amazon forest [57,58]. *Nyssorhynchus darlingi* is not always the dominant species in the *Nyssorhynchus* larval community that emerges with anthropogenic change: for example, in Mâncio Lima, Acre state, it is *Ny. albitarsis s.l.* [15]; in Labrea, Amazon state, *Ny. triannulatus* [24]; and in Pôrto Velho, Rondônia state, *Ny. braziliensis* [59]. However, *Ny. darlingi* may be the species that best adapts to human behavior in the Amazon region relative to vectorial capacity [49,60–62].

Our study was noteworthy for the micro-geographical analysis of larvae sampled, measuring different characteristics of vector ecology at sampling-point and fishpond levels. Anophelinae species L1 showed different behavior among the three sites: urban Site 1 had a decrease in August and September; however there was an increase in August in urban Site 2; whereas for rural Site 3, there was a decrease in larval counts in April and May. These results did not change in the multivariate analysis, in contrast with Anophelinae L2—L4, which present an inverse association in urban Site 1 after adjustment, indicating some influential cofactor that was not measured by this study.

Nevertheless, the primary vector *Ny. darlingi* L2—L4 group—identified to species level-shows a singular pattern: no difference of larval numbers in fishponds detected in urban Sites 1 and 2 during the rainy to dry season transition, in both univariate and multivariate analysis. There was also no seasonal difference for *Ny. darlingi* in early study [15], however, they incorporated a larger time frame (2 years) with larger intervals between larval sampling efforts (5–6 months). Interestingly, in our Site 3 (rural), there was a significant increase in larval numbers

between the February baseline and both August and September, months that correspond to the dry season. Possibly, fishponds play a more important role in the maintenance of *Ny. darlingi* during the transition from rainy to dry season in rural landscapes than our study demonstrates. Similar results were found in rural settlements with the presence of artificial breeding-sites [10,47].

Seasonal malaria is common in the Amazon region, associated with *Ny. darlingi* population density and rainfall patterns. In urban and suburban areas in Rondônia state [45], malaria increased at the end of the dry season and the beginning of the next rainy season in landscapes with mainly natural breeding-sites (riverside malaria); in contrast, in landscape dominated by artificial breeding-sites (so-called dryland malaria), both malaria and *Ny. darlingi* remain high throughout the year. Here, a simple linear correlation between monthly precipitation and *P. vivax* notifications showed a positive association for rainfall seasonality and malaria cases, mainly in 2016 (2016 $r = 0.75$; 2017 = 0.43), indicating some seasonal effect on malaria numbers (S1 Fig). However, these monthly notifications could have been more informative had they been adjusted for the appropriate landscape profile (urban/rural). For *Ny. darlingi* larvae, a major sampling effort with more sites in each landscape type in a multi-year survey is needed to confirm this seasonal pattern.

In urban Site 1, the increase of *Ny. darlingi* larvae in active fishponds, not detected for Anophelinae L1 and L2—L4 groups, supports the earlier study [15], demonstrating that economically active fishponds are important larval habitats for primary vectors. Emerging aquatic vegetation was strongly associated with all three Anophelinae groups in the overall regression, reinforcing the recommendation by WHO [63], that cleaning the margins can be an effective environmental control for Amazon *Nyssorhynchus* sp. The presence of *Culex* sp. species and egg rafts was constant in the survey, suggesting they share the same ecological niche as the Anophelinae L2—L4 group. Most *Culex* sp. were identified as subgenus *Melanoconion*, a group that contains species that are regional arbovirus vectors [64]. Thus, fish farming may open larval habitats for other Culicidae species of epidemiological importance. The presence of fish was common in the fishponds surveyed (even abandoned ones), showing that Anophelinae larval species readily coexist with the local fish community, or amphibians according to a microecological food web of aquatic habitat [65]. Prospects for putative biological control seem unclear in this case unless exotic larvivorous fish species were to be utilized, but they represent other risks for the local environment and are not a feasible option [66].

Water turbidity was slightly associated with Anophelinae numbers, with *Ny. darlingi* being found previously in turbid water [24]. We report a significant association with shaded or low light environments for the three Anophelinae groups, a feature associated previously with *Ny. darlingi* ecology [48]. High values of pH (>7) and nitrates (40 mg/L) appear to be limiting factors for the Anophelinae aquatic habitat. Although carbonated hardness (an alkalinity indicator), shows a similar pattern in the decrease of Anophelinae L1 and L2—L4 groups, for *Ny. darlingi* there was no significant association. More surprisingly, the increase in *Ny. darlingi* larvae in waters with dissolved chlorine suggested possible tolerance of immatures to polluted aquatic habitats. This was also detected for the Anophelinae L1, representing an important feature of opportunistic species that invade new aquatic niches in human occupation without environmental sanitation, and may be linked to phenotypic plasticity of ion regulation of Amazon mosquito Culicidae larvae under different physical-chemical conditions [67].

Aside from the non-identification of L1 larvae, mentioned above, a second limitation of this study was that we planned to measure the perimeter of each fishpond to test for an association with larval abundance [10]. We initially measured each fishpond but, due the high number of ponds (n = 63), it was not realistic to accurately measure change in water level in each one for each of the six months. Thirdly, there were some technical problems with measuring

instruments, resulting in gaps in some of the independent variables of the survey, reducing sampling effort. Fourthly, we intensively sampled two urban sites but only one rural one, mainly due to complex logistical issues. Finally, there is an important relationship between households with malaria incidence and distance to breeding sites for *Ny. darlingi* [48] and *P. vivax* infection [68] measurement of which was beyond the scope of our study.

Nevertheless, our study does provide important information about temporal variation and environment features of larvae of the primary vector *Ny. darlingi* at micro-spatial levels (sampling points of fishponds), as well as *Ny. darlingi* adult profiles in nearby households. Tailored LSM strategies accounting for this heterogeneity, such the use of biological larvicides [69], need to be routinely incorporated in malaria integrated control to reduce transmission in Mâncio Lima, and in other cities of Vale do Jurua region.

## Supporting information

**S1 Fig. Monthly distribution of reported malaria cases (*P. vivax* and *P. falciparum*) and accumulated monthly precipitation.** Mancio Lima municipality 2016 and 2017. Pearson correlations: in 2016 for *P. vivax* and precipitation r = 0.75; for *P. falciparum* and precipitation r = 0.51; in 2017 for *P. vivax* and precipitation r = 0.43; for *P. falciparum* and precipitation r = 0.47. (Source: Malaria: http://www2.datasus.gov.br/DATASUS; Precipitation: clima1.cptec.inpe.br).
(TIF)

**S1 Table. Independent variable and response options (by classification or instrument measure), level of subject analysis (sampling-point or fishpond), and sampling effort during 2017 according to the sampling schedule (6-month for Feb, Mar, Apr, May, Aug, Sept; 3-month for Feb, Mar, Apr; 3-month for May, Aug, Sept).**
(DOCX)

**S2 Table. Fishpond numbers by site and collection month in Mancio Lima, Acre, Brazil 2017.** Fishpond column: numbers of fishponds identified per respective month; values in brackets are total fishponds surveyed for the respective month. Dry column: fishponds with absence of water (by seasonality or owner management). Not surveyed column: fishponds unsampled (due to flooded fields or no possible access to property).
(DOCX)

**S3 Table. Environment independent variables by site in Mancio Lima, Acre, Brazil 2017.**
*fishpond level. **sampling-point level.
(DOCX)

## Acknowledgments

We thank Marcelo Urbano Ferreira and Pablo Secato Fontoura (Instituto de Ciências Biomédicas—Universidade do São Paulo); Francis de Melo Santos (Gerente de Endemias de Mâncio Lima); Rodrigo Medeiros de Souza (Laboratório de Doenças Infecciosas na Amazônia Ocidental—Universidade Federal do Acre); Vera Lucia Carvalho da Silva (Ministério da Saúde do Brasil); Reginaldo Grenzi da Silva (Biblioteca, FSP-USP); the residents of Mâncio Lima and Cruzeio do Sul (AC) for their hospitality; and the Environmental Systems Research Institute, Inc. (Esri) for permission to use the satellite images.

## Author Contributions

**Conceptualization:** Paulo Rufalco-Moutinho, Jan E. Conn, Paulo Eduardo Martins Ribolla.

**Data curation:** Paulo Rufalco-Moutinho, Gabriel Carrasco-Escobar.

**Formal analysis:** Paulo Rufalco-Moutinho, Jan E. Conn.

**Funding acquisition:** Dionicia Gamboa, Joseph M. Vinetz, Maria Anice Mureb Sallum, Jan E. Conn, Paulo Eduardo Martins Ribolla.

**Investigation:** Paulo Rufalco-Moutinho, Samir Moura Kadri, Diego Peres Alonso, Marta Moreno, Gabriel Carrasco-Escobar, Jan E. Conn, Paulo Eduardo Martins Ribolla.

**Methodology:** Paulo Rufalco-Moutinho.

**Project administration:** Dionicia Gamboa, Joseph M. Vinetz, Jan E. Conn, Paulo Eduardo Martins Ribolla.

**Supervision:** Jan E. Conn, Paulo Eduardo Martins Ribolla.

**Visualization:** Paulo Rufalco-Moutinho.

**Writing – original draft:** Paulo Rufalco-Moutinho.

**Writing – review & editing:** Paulo Rufalco-Moutinho, Samir Moura Kadri, Marta Moreno, Catharine Prussing, Dionicia Gamboa, Joseph M. Vinetz, Maria Anice Mureb Sallum, Jan E. Conn, Paulo Eduardo Martins Ribolla.

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
