## [Editor Report · Decision Letter 0]

21 Jan 2021

PONE-D-21-00873

Population dynamics of the primary malaria vector Nyssorhynchus darlingi in a high transmission setting dominated by fish farming in western Amazonian Brazil

PLOS ONE

Dear Dr. Rufalco-Moutinho,

Thank you for submitting your manuscript to PLOS ONE.Please fix the first sentence of the abstract.

We look forward to receiving your revised manuscript.

Kind regards,

Olle Terenius

Academic Editor

PLOS ONE

Journal Requirements:

2. We note that Figures 1 and 2 in your submission contain map/satellite images which may be copyrighted. All PLOS content is published under the Creative Commons Attribution License (CC BY 4.0), which means that the manuscript, images, and Supporting Information files will be freely available online, and any third party is permitted to access, download, copy, distribute, and use these materials in any way, even commercially, with proper attribution. For these reasons, we cannot publish previously copyrighted maps or satellite images created using proprietary data, such as Google software (Google Maps, Street View, and Earth). For more information, see our copyright guidelines: http://journals.plos.org/plosone/s/licenses-and-copyright.

2.1.    You may seek permission from the original copyright holder of Figures 1 and 2 to publish the content specifically under the CC BY 4.0 license. 

2.2.    If you are unable to obtain permission from the original copyright holder to publish these figures under the CC BY 4.0 license or if the copyright holder’s requirements are incompatible with the CC BY 4.0 license, please either i) remove the figure or ii) supply a replacement figure that complies with the CC BY 4.0 license. Please check copyright information on all replacement figures and update the figure caption with source information. If applicable, please specify in the figure caption text when a figure is similar but not identical to the original image and is therefore for illustrative purposes only.

---

## [Author Response · Author response to Decision Letter 0]

1 Mar 2021

To the PLOS One Editors,

Below are our responses to the reviewers of the manuscript we submitted: [PONE-D-21-00873] - [EMID:61ecbed848a59a53], title: Population dynamics of the primary malaria vector Nyssorhynchus darlingi in a high transmission setting dominated by fish farming in western Amazonian Brazil. 

RESPONSE: We modified the text etc. according to the style requirements, including the file names. Minor changes were made in this new version: we chose a new title, to more clearly differentiate this manuscript from previous papers by our group. The Abstract was re-written to in accordance to PLOS recommendations for word count. Author were specified in the “Contributor Roles”, in Authors section on “Manuscript Data” topic in the Editorial Manager.

2. We note that Figures 1 and 2 in your submission contain map/satellite images which may be copyrighted. All PLOS content is published under the Creative Commons Attribution License (CC BY 4.0), which means that the manuscript, images, and Supporting Information files will be freely available online, and any third party is permitted to access, download, copy, distribute, and use these materials in any way, even commercially, with proper attribution. For these reasons, we cannot publish previously copyrighted maps or satellite images created using proprietary data, such as Google software (Google Maps, Street View, and Earth). For more information, see our copyright guidelines: http://journals.plos.org/plosone/s/licenses-and-copyright.

RESPONSE: We contacted the ESRI company and received permission to use the copyright figures. The new source of figures was the “ArcGIS Online USGS Landsat Baselayer”, and map items were inserted using Microsoft Power Point. The copyright permission request document was provide by ESRI from an owner form (PDF file, attached here), and not the PLOS form model CC BY 4.0, although this permission contemplates full use of these figures, according to the PLOS's requirements about copyright issues. 

Yours truly,

Paulo Rufalco-Moutinho

For the authors.

---

## [Editor Report · Decision Letter 1]

24 Mar 2021

Ecology and larval population dynamics of the primary malaria vector Nyssorhynchus darlingi in a high transmission setting dominated by fish farming in western Amazonian Brazil

PONE-D-21-00873R1

Dear Dr. Rufalco-Moutinho,

We’re pleased to inform you that your manuscript has been judged scientifically suitable for publication and will be formally accepted for publication once it meets all outstanding technical requirements.

Kind regards,

Olle Terenius

Academic Editor

PLOS ONE
---

## [Editor Report · Acceptance letter]

29 Mar 2021

PONE-D-21-00873R1 

Ecology and larval population dynamics of the primary malaria vector *Nyssorhynchus darlingi* in a high transmission setting dominated by fish farming in western Amazonian Brazil 

Dear Dr. Rufalco-Moutinho:

I'm pleased to inform you that your manuscript has been deemed suitable for publication in PLOS ONE. Congratulations! Your manuscript is now with our production department. 

Kind regards, 

on behalf of

Dr. Olle Terenius 

Academic Editor

PLOS ONE